# Hierarchical Prototype Learning for Semantic Segmentation

**Seoha Lim**[1]*, **Jinmyeong Kim**[2]*, **Jieun Kim**[1], **Sung-Bae Cho**[2]
[1]Department of Artificial Intelligence, Yonsei University
[2]Department of Computer Science, Yonsei University
{seoha815,jmkim_,lilly9928,sbcho}@yonsei.ac.kr

## Abstract

Conventional semantic segmentation methods often fail to distinguish fine-grained parts within the same object because of missing links between part-level cues and object-level semantics. Inspired by how humans recognize objects, which involves first identifying them as a whole and then distinguishing their parts, we propose a hierarchical prototype-based segmentation method called Hierarchical Prototype Segmentation (*HiPoSeg*). This builds a structured prototype space that captures both abstract object-level representations and detailed part-level features, enabling consistent alignment between levels. *HiPoSeg* leverages a hierarchical contrastive learning strategy to structure semantic representations across levels, encouraging both intra-level discrimination and cross-level consistency. Experiments on standard benchmarks such as Cityscapes, ADE20K, Mapillary Vistas 2.0, and PASCAL-Part-108 demonstrate that *HiPoSeg* produces consistent performance improvement with an average gain of +3.07%p mIoU without any additional inference cost.

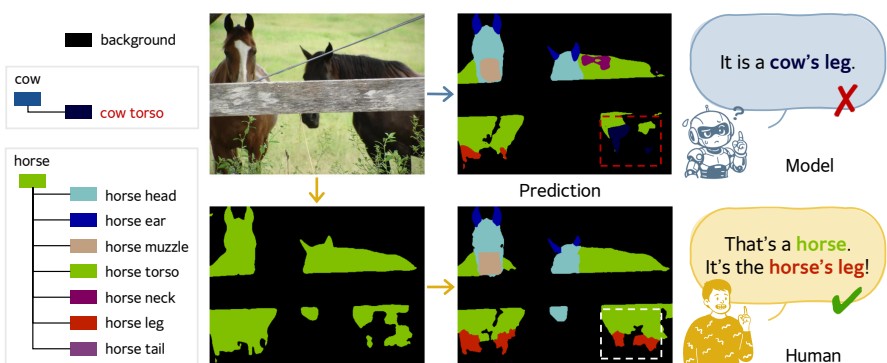

Figure 1: Comparison of semantic segmentation between human and computational model.

## 1 Introduction

Semantic segmentation, a fundamental task in computer vision, requires assigning a semantic label to every pixel of an image (Xie et al., 2021; Li et al., 2022; 2023; Fu et al., 2019; Zhao et al., 2019; Sung et al., 2024; Zhou et al., 2022). Although deep learning has significantly advanced this field (Kim & Cho, 2022), most current approaches still treat it as a flat classification problem, where each pixel is assigned directly to a predefined class label without considering any structural relationships among classes (Michieli et al., 2020; Chen et al., 2020; Huang et al., 2019; Chen et al., 2018; Zhao et al., 2018). This lack of structure can result in semantic inconsistencies, especially when visually similar parts belong to different classes. In contrast, the human visual system approaches recognition quite differently: it operates hierarchically. For example, in Fig. 1, when we observe an

---

*Equal contribution.

object, we first recognize its broader category (e.g., *horse*) and then identify its finer components (e.g., *horse's leg*). This top-down process allows for faster and more accurate recognition by reducing the scope of plausible interpretations: once we know it as a horse, we naturally expect its parts to be horse-related. Rather than using such hierarchical priors, conventional segmentation models treat all classes as independent, limiting their ability to reason contextually or semantically. This limitation is especially problematic in fine-grained segmentation tasks or scenarios involving a large number of visually similar categories. Without hierarchical guidance, models often fail to disambiguate small object parts or rare classes, resulting in degraded performance and a lack of structural coherence.

To overcome these limitations, we propose *HiPoSeg*, a novel segmentation method inspired by human visual perception, which is built on two key mechanisms. 1) Hierarchical prototype learning: Instead of learning pixel-to-class mappings in isolation, we organize class representations hierarchically and learn prototypes at each semantic level. This allows the model to perform recognition in a top-down fashion: first estimating coarse-level categories and then refining to fine-grained sub-categories conditioned on the higher-level predictions. This structure encodes the dependency of categories and promotes consistency within semantic groups. 2) Multi-level class alignment constraint: To ensure coherence between the predicted levels, we introduce a constraint that enforces alignment between hierarchical layers. Once a high-level category is predicted, lower-level predictions are guided to remain within its semantic subtree. This significantly reduces misclassification among semantically unrelated classes and improves part-level accuracy. By integrating these two mechanisms, *HiPoSeg* mimics the hierarchical nature of human perception, offering a structured alternative to flat segmentation. This not only improves prediction accuracy, but also enhances semantic consistency, especially in complex scenes with fine-grained parts and large category spaces.

In sum, this paper has the following contributions.

- **Hierarchical prototype learning**: We maintain higher- and lower-level class prototypes aligned with the label tree and train embeddings from multiple feature stages to be organized around these prototypes. This encourages parts of the same object to share global context while stably differentiating fine-grained representations, thereby achieving semantic consistency across the high–low hierarchy.

- **Multi-level class alignment constraint**: We introduce a hierarchy-aware alignment constraint that suppresses probability leakage and representation drift toward non-subcategories when a high-level category is activated. This improves fine-grained stability and reduces misclassification in boundary and confusing regions.

- **Training-only, plug-and-play efficiency**: The proposed method is minimally attached to the output end of existing segmentation models as an auxiliary training component. It is used only during training and removed at inference, introducing no additional parameters, computation, or latency at the test time, and it integrates easily with various backbones and decoders.

## 2 RELATED WORK

### 2.1 HUMAN VISUAL COGNITION IN DEEP LEARNING

Human visual cognition is often characterized as a hierarchical process that quickly anchors a coarse (global) category and then progressively refines it into parts (Bar, 2003; Bar et al., 2006; Bar, 2004). This top-down mechanism narrows the candidate space early and regularizes fine decisions with global context, improving efficiency and robustness (Bar, 2003). Similar ideas have been exploited in deep scene understanding, for example, establishing global context and then refining local details, or using a coarse map to guide boundary correction (Ji et al., 2020; Meletis & Dubbelman, 2018). However, many previous methods implicitly assume a flat label space at the level of network design or loss construction, and thus do not explicitly model hierarchical constraints.

We operationalize these cognitive insights via a hierarchical prototype representation that is used only during training. We first stabilize global semantics at the high level and then separate fine categories at the low level in a contrastive manner, while enforcing an alignment constraint to maintain consistency across levels (Li et al., 2022). In practice, we implement the coarse to fine process

through a combination of prototype memory, contrastive objectives, and margin-based alignment, thus achieving both object-level context and part-level discrimination (Bar, 2009). Importantly, the proposed method is training-only (zero inference overhead), making it a plug-and-play addition to standard backbones and decoders.

## 2.2 HIERARCHICAL SEMANTIC SEGMENTATION

A long line of work has sought to exploit hierarchy in semantic segmentation, either by structurally integrating global and local context or by reporting performance across high-level/low-level label groups. For example, OCRNet (Yuan et al., 2020) strengthens object-region representations to encode global context, while RegionSeg (Hu et al., 2021) leverages regional partitioning and relations to improve mask quality. More direct treatments of hierarchy introduce information flow or constraints between label levels (e.g., HSSN (Li et al., 2022) and LogicSeg (Li et al., 2023)). They highlight the need for hierarchical reasoning, but combine signals primarily at the probability/logit level or handle hierarchy as a fixed auxiliary term, falling short of structuring the class representation itself.

We lift hierarchy from a mere loss add-on to a design principle of the representation space. Concretely, we maintain class-wise prototype memories at both the high- and low-levels and define/update each prototype as the average of normalized features with the same label (Snell et al., 2017) (Sec. 3.1). We then apply hierarchical contrastive learning (Sec. 3.2): (a) feature-prototype and prototype-prototype objectives at the high level, (b) their counterparts at the low level, and (c) a high low-level margin alignment that preserves the topology across levels. This design induces a hierarchically organized embedding space without relying on hand-crafted fusion rules.

## 2.3 CONTRASTIVE LEARNING FOR SEMANTIC SEGMENTATION

Contrastive learning has been widely adopted to strengthen pixel/region representations in segmentation (Chen et al., 2020; Moon & Cho, 2025). Representative methods such as ContrastSeg (Wang et al., 2021) construct positive/negative pairs at the pixel or region level to widen decision margins, while RegionSeg (Hu et al., 2021) performs region-level contrast to enhance mid-level features. Prototype-based variants stabilize training by anchoring features to class-centric representatives (Zhou et al., 2022; Kee et al., 2024), and recent work further accelerates convergence via context-aware sampling and hybrid objectives (Sung et al., 2024).

However, most contrastive methods assume a flat label space and, therefore, overlook hierarchical relations when defining positives and negatives (Tang et al., 2023). We address this limitation through a hierarchical prototype bank combined with hierarchy-aware margin alignment. Concretely, (i) the high- and low-feature contrasts pull the embeddings toward the correct prototype at each level; (ii) the contrasts between the high- and low-level classes enforce clear inter-class separation within each level; and (iii) an alignment constraint restricts the high- and low-level distances to $(0, \text{margin})$ while ensuring that distances among the high-level classes remain above the margin. This design naturally induces a coarse-to-fine curriculum: the high-level space first narrows down candidate classes, and then the low-level space performs fine-grained discrimination. The alignment term is particularly important, as it prevents gradient interference and prototype drift that can arise when multi-level contrastive losses are naively combined, while maintaining the training-only, zero-overhead property at inference.

## 3 PROPOSED METHOD

As illustrated in Fig.2, a new method is proposed for hierarchical class-wise prototype learning for semantic segmentation. Before learning the prototype, Sec.3.1 describes the construction of the hierarchical prototype space, and Sec.3.2 elaborates details of the prototype learning procedure, where prototype-based hierarchical contrastive learning is employed.

## 3.1 HIERARCHICAL PROTOTYPE SPACE CONSTRUCTION

In this section, we define a prototype space that reflects the hierarchical label structure and describe how to update it stably during training. Let the dataset be $\mathcal{D} = \{(X_k, Y_k)\}_{k=1}^{K}$, and $\omega_k$ denote the

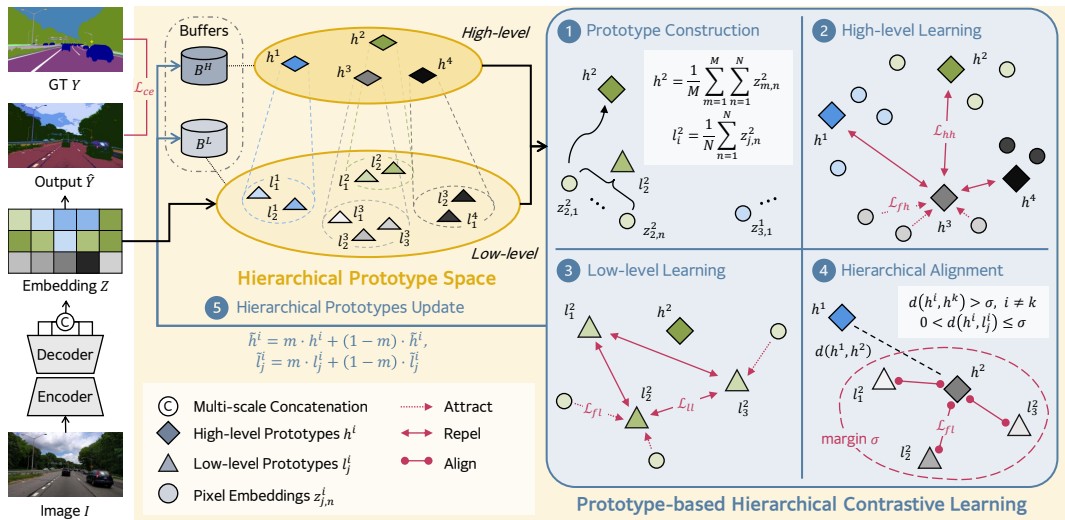

Figure 2: Overview of *HiPoSeg*, composed of (1) prototype construction: pixel features are grouped by low-level labels and their corresponding high-level mappings to initialize both low- and high-level prototypes, (2) high-level learning: pixel features are pulled toward their high-level prototypes that are enforced to be mutually separated, (3) low-level learning: after high-level convergence, low-level discrimination is refined using the same pull–push mechanism, and (4) hierarchical alignment: each low-level prototype is constrained to remain close to its parent high-level prototype while preserving separation between different high-level groups.

set of pixel indexes of the image $X_k$. Given an encoder–decoder segmentation model $f_\theta$ (backbone + decoder), the pixel embeddings are

$$Z_k \;=\; \{\, z_k^n \in \mathbb{R}^d \mid n \in \omega_k \,\} \;=\; f_\theta(X_k). \tag{1}$$

Assume a two-level hierarchy of labels (high-level and low-level). Let $Y_k = \{\, y_k^n \mid n \in \omega_k \,\}$ be the low-level labels of $X_k$, where $y_k^n$ is the label corresponding to $z_k^n$. We obtain the high-level labels by a fixed mapping $\pi : \mathcal{Y}^L \to \mathcal{Y}^H$ as

$$y_k'^n \;=\; \pi\big(y_k^n\big), \qquad Y_k' \;=\; \{\, y_k'^n \mid n \in \omega_k \,\}. \tag{2}$$

For prototype construction and contrastive learning, we use a projection head $g_\phi : \mathbb{R}^d \to \mathbb{R}^{d'}$ followed by $\ell_2$-normalization:

$$\tilde{z}_k^n \;=\; \frac{g_\phi(z_k^n)}{\big\| g_\phi(z_k^n) \big\|_2} \;\in\; \mathbb{R}^{d'}. \tag{3}$$

We maintain two memory buffers for prototypes in each hierarchy: a high-level buffer $\mathcal{M}^H$ and a low-level buffer $\mathcal{M}^L$,

$$\mathcal{M}^H \;=\; \big[\, h^i \,\big]_{i \in \mathcal{Y}^H}, \qquad \mathcal{M}^L \;=\; \big[\, l_j^i \,\big]_{i \in \mathcal{Y}^H,\, j \in \mathcal{Y}^L(i)}, \tag{4}$$

where $\mathcal{Y}^L(i) = \{\, j \in \mathcal{Y}^L \mid \pi(j) = i \,\}$ is the set of low-level parts of the high-level class $i$.

We compute batch-wise (or running) prototype estimates using normalized embeddings $\tilde{z}_k^n$ as

$$\mu^i \;=\; \frac{\displaystyle\sum_k \sum_{n \in \omega_k} \mathbb{1}\big[\, y_k'^n = i \,\big]\, \tilde{z}_k^n}{\displaystyle\sum_k \sum_{n \in \omega_k} \mathbb{1}\big[\, y_k'^n = i \,\big] \;+\; \epsilon}, \qquad \mu_j^i \;=\; \frac{\displaystyle\sum_k \sum_{n \in \omega_k} \mathbb{1}\big[\, y_k^n = j \,\big]\, \tilde{z}_k^n}{\displaystyle\sum_k \sum_{n \in \omega_k} \mathbb{1}\big[\, y_k^n = j \,\big] \;+\; \epsilon}, \tag{5}$$

where $\mathbb{1}[\cdot]$ is the indicator function and $\epsilon > 0$ is a small constant for numerical stability. The prototypes are then $\ell_2$-normalized:

$$h^i \;=\; \frac{\mu^i}{\|\mu^i\|_2}, \qquad l_j^i \;=\; \frac{\mu_j^i}{\|\mu_j^i\|_2}. \tag{6}$$

## 3.2 Prototype-based Hierarchical Contrastive Learning

This section describes (i) contrastive learning at the *high-level*, (ii) contrastive learning at the *low-level*, (iii) an alignment constraint that couples the *two levels*, and (iv) *prototype memory* updates and training schedule. All similarities are derived from the Euclidean metric. Let $\delta(a,b) = \|a-b\|_2$ be the Euclidean distance and $s(a,b) = -\delta(a,b)^2$ its negative (used inside softmax).

**High-level Contrastive Learning.** Motivated by a coarse-to-fine recognition process, we first learn broad semantics by contrasting pixel features against *high-level* prototypes and also separating *high-level* prototypes from each other. For a pixel $z_k^n$ with *high-level* ground-truth $y'^* \in \mathcal{Y}^H$, let the positive prototype be $p_{y'^*}^h = h^{y'^*}$ and negatives $\{p_{y'}^h = h^{y'}\}_{y' \in \mathcal{Y}^H, y' \neq y'^*}$. The feature–prototype contrastive loss is

$$\mathcal{L}_{fh} = -\frac{1}{|\mathcal{N}^*|} \sum_{k,n} \log \frac{\exp\big(s(\tilde{z}_k^n, p_{y'^*}^h)/\tau\big)}{\sum_{y' \in \mathcal{Y}^H} \exp\big(s(\tilde{z}_k^n, p_{y'}^h)/\tau\big)}, \tag{7}$$

where $\tau > 0$ is the temperature and $|\mathcal{N}^*|$ is the number of valid pixels in the batch. Currently, we discourage *high-level* prototypes from collapsing via a repulsive regularizer.

$$\mathcal{L}_{hh} = \frac{1}{|\mathcal{Y}^H|(|\mathcal{Y}^H| - 1)} \sum_{\substack{i,k \in \mathcal{Y}^H \\ i \neq k}} \exp\big(s(h^i, h^k)/\kappa\big), \tag{8}$$

with temperature $\kappa > 0$. Minimizing $\mathcal{L}_{fh} + \mathcal{L}_{hh}$ pulls the features towards their correct *high-level* prototypes, pushes them away from the incorrect ones, and separates the prototypes themselves.

**Low-level Contrastive Learning.** After establishing high-level semantics, we learn fine-grained semantics by contrasting features and *low-level* prototypes, together with a separation term among *low-level* prototypes. For a pixel $z_k^n$ with *low-level* ground-truth $y^* \in \mathcal{Y}^L$, thel loss of feature–prototype is

$$\mathcal{L}_{fl} = -\frac{1}{|\mathcal{N}^*|} \sum_{k,n} \log \frac{\exp\big(s(\tilde{z}_k^n, p_{y^*}^l)/\tau\big)}{\sum_{y \in \mathcal{Y}^L} \exp\big(s(\tilde{z}_k^n, p_y^l)/\tau\big)}, \tag{9}$$

and the prototype–prototype repulsion is

$$\mathcal{L}_{ll} = \frac{1}{|\mathcal{Y}^L|(|\mathcal{Y}^L| - 1)} \sum_{\substack{j,k \in \mathcal{Y}^L \\ j \neq k}} \exp\big(s(l^j, l^k)/\kappa\big). \tag{10}$$

These terms make features close to their correct *low-level* prototypes and keep distinct *low-level* prototypes well separated.

**High–low Alignment Constraint.** Using only the above losses may cause confusion between high-level and low-level semantics. We therefore align low-level prototypes with their high-level prototypes while enforcing *inter-high* separation. Let $l_j^i$ be a low-level prototype of the high-level class $h^i$. We constrain *low-to-high* proximity via

$$\mathcal{L}_{\text{align\_in}} = \frac{1}{|\mathcal{Y}^L|} \sum_{(i,j):\,\pi(j)=i} \max\big(0, \delta(l_j^i, h^i) - \sigma_1\big), \tag{11}$$

and keep different high-levels well separated via

$$\mathcal{L}_{\text{align\_out}} = \frac{1}{|\mathcal{Y}^H|(|\mathcal{Y}^H| - 1)} \sum_{\substack{i,k \in \mathcal{Y}^H \\ i \neq k}} \max\big(0, \sigma_2 - \delta(h^i, h^k)\big). \tag{12}$$

The total alignment loss is

$$\mathcal{L}_{\text{align}} = \alpha\, \mathcal{L}_{\text{align\_in}} + \beta\, \mathcal{L}_{\text{align\_out}}, \tag{13}$$

with margins $0 < \sigma_1 < \sigma_2$ (we use $\sigma_1=0.25$, $\sigma_2=1$) and weights $\alpha, \beta > 0$.

The total training loss is a weighted sum of all components:

$$\mathcal{L} = \lambda_1\, \mathcal{L}_{\text{ce}} + \lambda_2\, \mathcal{L}_{fh} + \lambda_3\, \mathcal{L}_{hh} + \lambda_4\, \mathcal{L}_{fl} + \lambda_5\, \mathcal{L}_{ll} + \lambda_6\, \mathcal{L}_{\text{align}}. \tag{14}$$

**Prototype Memory Update and Training Schedule.** After computing batch-wise prototypes, the memory bank is updated by momentum. Let $\hat{h}^i$ and $\hat{l}_j^i$ be the newly estimated (normalized) prototypes from the current batch, and $h^i$, $l_j^i$ denote the entries stored in the memory:

$$h^i \leftarrow m\,h^i + (1-m)\,\hat{h}^i, \qquad l_j^i \leftarrow m\,l_j^i + (1-m)\,\hat{l}_j^i, \tag{15}$$

with momentum $m \in (0,1)$ (we use $m=0.9$).

To stabilize the representation and follow a top–down curriculum, we employ a phased schedule: (i) for the first 7.5% of total iterations, learn feature representations only (no prototype learning); (ii) after 7.5%, enable high-level prototype learning; (iii) after 22.5%, enable low-level prototype learning as well; and (iv) after 37.5%, additionally activate hierarchical alignment losses. This schedule first consolidates coarse semantics and then refines fine-grained semantics under explicit high–low alignment.

# 4 EXPERIMENTAL RESULTS

## 4.1 EXPERIMENTAL SETUP

**Datasets.** We validate our method on four benchmark datasets, such as Cityscapes (Cordts et al., 2016), ADE20K (Zhou et al., 2016), Mapillary Vistas 2.0 (Neuhold et al., 2017), and PASCAL-Part-108 (Michieli et al., 2020), using their hierarchical label versions. Dataset statistics and hierarchy definitions are followed by (Li et al., 2023).

- **Cityscapes.** Urban street-scene dataset (2,975/500/1,524 train/val/test). Two-label levels: 7 coarse classes and 19 fine classes.

- **ADE20K.** Daily-scene dataset (20,210/2,000/3,000 train/val/test). Three-label levels: 3 top-level, 14 mid-level, and 150 fine-grained classes.

- **Mapillary Vistas 2.0.** Urban street-scene dataset (18,000/2,000/5,000 train/val/test). Three-label levels: 4 top-level, 16 mid-level, and 124 fine-grained classes.

- **PASCAL-Part-108.** Object part parsing dataset (4,998/5,105 train/test). Two-label levels: 21 coarse classes and 108 part-level classes.

Although the original hierarchies are defined with either two or three label levels depending on the dataset, our method does not rely on the absolute number of levels but only requires the availability of a higher-level concept. Therefore, for consistency, we unify all datasets into a two-level hierarchy in our experiments; specifically, for ADE20K and Mapillary Vistas 2.0, we use the mid-level as the higher-level (coarse) concept and the fine-grained labels as the lower-level classes.

We adopt the standard mean Intersection-over-Union (mIoU) for semantic segmentation. For hierarchy-aware evaluation, we report $\text{mIoU}^\ell$ at each label level $\ell$, where a larger $\ell$ denotes a higher (coarser) level in the hierarchy. We train DeepLabV3+ (Chen et al., 2018) with a ResNet-101 backbone using SGD (lr=1e-2, momentum 0.9, weight decay 1e-4). For Cityscapes and Mapillary Vistas 2.0, we use $512 \times 1024$ crops, batch size 8, and train for 80K iterations. For ADE20K and PASCAL-Part-108, we use $512 \times 512$ crops, batch size 16, and train for 60K iterations.

## 4.2 QUANTITATIVE EVALUATION

**Cityscapes.** Tab. 1 shows that integrating our method into DeepLabV3+ improves mIoU by +10.49%p over the baseline and achieves an average gain +3.07%p compared to competing methods. In particular, *HiPoSeg* surpasses hierarchical-semantics approaches such as HSSN and LogicSeg by +1.02%p and +0.84%p, respectively. In addition, it records the highest mIoU (84.04%) among all methods, outperforming both CNN-based designs (e.g., PSPNet and PSANet) and recent prototype-driven approaches (e.g., ProtoSeg and ContextSeg). Notably, while many competitors exploit stronger backbones such as HRNet-W48, *HiPoSeg* achieves superior results with ResNet-101, highlighting its efficiency and robustness. These results validate the effectiveness of hierarchical prototype contrastive learning on Cityscapes.

Table 1: Quantitative results on the Cityscapes `val` set.

| Method | Backbone | mIoU (%) |
|---|---|---|
| PSPNet (Zhao et al., 2017) | ResNet-101 | 80.91 |
| PSANet (Zhao et al., 2018) | ResNet-101 | 80.96 |
| PAN (Li et al., 2018) | ResNet-101 | 81.12 |
| Acfnet (Zhang et al., 2019) | ResNet-101 | 81.60 |
| DANet (Fu et al., 2019) | ResNet-101 | 81.52 |
| CCNet (Huang et al., 2019) | ResNet-101 | 81.08 |
| OCRNet (Yuan et al., 2020) | ResNet-101 | 82.33 |
| ContrastSeg (Wang et al., 2021) | ResNet-101 | 79.20 |
| ContrastSeg (Wang et al., 2021) | HRNet-W48 | 81.40 |
| RegionSeg (Hu et al., 2021) | ResNet-101 | 81.30 |
| RegionSeg (Hu et al., 2021) | HRNet-W48 | 81.90 |
| Multi-scale (Pissas et al., 2022) | ResNet-101 | 79.00 |
| Multi-scale (Pissas et al., 2022) | HRNet-W48 | 79.59 |
| ProtoSeg (Zhou et al., 2022) | HRNet-W48 | 81.10 |
| HSSN (Li et al., 2022) | ResNet-101 | 83.02 |
| LogicSeg (Li et al., 2023) | ResNet-101 | 83.20 |
| Contextrast (Sung et al., 2024) | HRNet-W48 | 82.20 |
| DeepLabV3+ (Chen et al., 2018) | ResNet-101 | 73.55 |
| **HiPoSeg** | ResNet-101 | **84.04** (+10.49) |

Table 2: Quantitative results on the ADE20K `val` set.

| Method | Backbone | mIoU (%) |
|---|---|---|
| OCRNet (Yuan et al., 2020) | HRNet-W48 | 44.92 |
| RegionSeg (Hu et al., 2021) | ResNet-101 | 46.85 |
| Multi-scale (Pissas et al., 2022) | ResNet-101 | 45.60 |
| Multi-scale (Pissas et al., 2022) | HRNet-W48 | 47.40 |
| ProtoSeg (Zhou et al., 2022) | HRNet-W48 | 43.00 |
| HSSN (Li et al., 2022) | ResNet-101 | 47.69 |
| LogicSeg (Li et al., 2023) | ResNet-101 | 48.46 |
| Contextrast (Sung et al., 2024) | HRNet-W48 | 43.42 |
| DeepLabV3+ (Chen et al., 2018) | ResNet-101 | 44.48 |
| **HiPoSeg** | ResNet-101 | **48.99** (+4.51) |

Table 3: Quantitative results on the Mapillary Vistas 2.0 `val` set.

| Method | Backbone | mIoU (%) |
|---|---|---|
| Seamless (Porzi et al., 2019) | ResNet-101 | 38.17 |
| HMSANet (Wang et al., 2020) | HRNet-W48 | 39.53 |
| OCRNet (Yuan et al., 2020) | HRNet-W48 | 38.26 |
| MaskFormer (Cheng et al., 2021) | ResNet-101 | 39.60 |
| HSSN (Li et al., 2022) | ResNet-101 | 40.16 |
| LogicSeg (Li et al., 2023) | ResNet-101 | 40.72 |
| DeepLabV3+ (Chen et al., 2018) | ResNet-101 | 31.65 |
| **HiPoSeg** | ResNet-101 | **41.42** (+9.77) |

Table 4: Quantitative results on the PASCAL-Part-108 `test` set.

| Method | Backbone | mIoU (%) |
|---|---|---|
| FCN-8s (Shelhamer et al., 2015) | ResNet-101 | 38.62 |
| SegNet (Badrinarayanan et al., 2015) | ResNet-101 | 36.42 |
| BSANet (Zhao et al., 2019) | ResNet-101 | 47.36 |
| GMNet (Michieli et al., 2020) | ResNet-101 | 47.21 |
| FLOAT (Singh et al., 2022) | ResNet-101 | 48.08 |
| HSSN (Li et al., 2022) | ResNet-101 | 48.32 |
| LogicSeg (Li et al., 2023) | ResNet-101 | 48.46 |
| DeepLabV3+ (Chen et al., 2018) | ResNet-101 | 46.90 |
| **HiPoSeg** | ResNet-101 | **49.33** (+2.43) |

Table 5: Ablation study of loss components on the Cityscapes `val` set.

| $L_{ce}$ | $L_{fh}$ | $L_{hh}$ | $L_{fl}$ | $L_{ll}$ | $L_{align}$ | mIoU (%) |
|---|---|---|---|---|---|---|
| ✓ | | | | | | 73.55 |
| ✓ | ✓ | ✓ | | | | 80.96 |
| ✓ | | | ✓ | ✓ | | 81.04 |
| ✓ | ✓ | ✓ | ✓ | ✓ | | 79.16 |
| ✓ | ✓ | ✓ | ✓ | ✓ | ✓ | **84.04** |

Table 6: Ablation study on hierarchical prototypes (mIoU %).

| High-level class | Low-level class | Cityscapes | ADE20K | Mapillary | PASCAL-Part-108 |
|---|---|---|---|---|---|
| | | 73.55 | 44.48 | 31.65 | 46.90 |
| | ✓ | 81.04 | 46.92 | 37.21 | 47.02 |
| ✓ | | 80.96 | 46.87 | 38.85 | 47.59 |
| ✓ | ✓ | **84.04** | **48.99** | **41.42** | **49.33** |

**ADE20K.** In Tab. 2, our method improves DeepLabV3+ by +4.51%p mIoU and outperforms alternatives by an average of +2.78%p. Compared to HSSN and LogicSeg, *HiPoSeg* achieves gains of +1.02%p and +0.84%p, respectively. Achieving the highest accuracy (48.99%) among all methods, *HiPoSeg* consistently establishes new state-of-the-art results under fair backbone settings. Importantly, its robustness is further demonstrated by outperforming several methods that rely on stronger backbones, confirming both the efficiency and scalability of the proposed method.

**Mapillary Vistas 2.0.** Tab. 3 shows that *HiPoSeg* yields an improvement of +9.77%p over the baseline and an average advantage of +2.01%p over competitors. Compared to HSSN and LogicSeg, the gains are +1.26%p and +0.70%p, respectively. *HiPoSeg* reaches the highest accuracy (41.42%) using ResNet-101, surpassing or matching methods. This demonstrates the scalability and robustness of hierarchical prototype learning on diverse large-scale street-scene datasets, where fine-grained distinctions are particularly challenging.

Table 7: High-level IoU comparison between baseline and *HiPoSeg* on the Cityscapes val set.

| high class | baseline (%) | HiPoSeg (%) | gap (%p) |
|---|---|---|---|
| flat | 96.54 | **96.90** | +0.36 |
| construction | 89.75 | **90.88** | +1.13 |
| object | 72.06 | **74.31** | +2.25 |
| nature | 90.53 | **91.48** | +0.95 |
| sky | 95.02 | **95.48** | +0.46 |
| human | 80.25 | **82.61** | +2.36 |
| vehicle | 87.06 | **93.49** | +6.43 |

Table 8: Low-level IoU comparison between baseline and *HiPoSeg* on the Cityscapes val set.

| low class | baseline (%) | HiPoSeg (%) | gap (%p) |
|---|---|---|---|
| road | 98.10 | **98.44** | +0.34 |
| sidewalk | 85.15 | **86.87** | +1.72 |
| building | 91.76 | **93.41** | +1.65 |
| wall | 40.06 | **52.54** | +12.48 |
| fence | 55.33 | **62.49** | +7.16 |
| pole | 66.21 | **70.85** | +4.64 |
| traffic light | 68.63 | **76.48** | +7.85 |
| traffic sign | 78.30 | **81.79** | +3.49 |
| vegetation | 92.09 | **92.96** | +0.89 |
| terrain | 57.44 | **63.05** | +5.61 |
| sky | 94.52 | **95.48** | +0.96 |
| person | 81.01 | **85.19** | +4.18 |
| rider | 57.83 | **67.84** | +10.01 |
| car | 92.88 | **96.14** | +3.26 |
| truck | 54.15 | **84.05** | +29.90 |
| bus | 69.93 | **89.83** | +19.90 |
| train | 23.88 | **83.47** | +59.59 |
| motorcycle | 58.76 | **73.02** | +14.26 |
| bicycle | 75.91 | **80.92** | +5.01 |

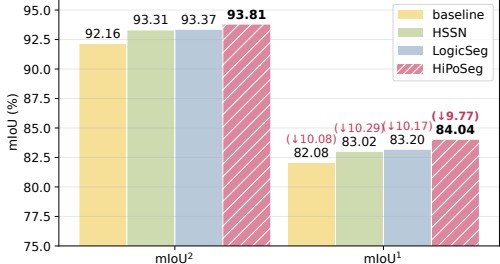

Figure 3: Comparison of $\text{mIoU}^1$ and $\text{mIoU}^2$ on Cityscapes ($\text{mIoU}^1$ for the low level, $\text{mIoU}^2$ for the high level).

**PASCAL-Part-108.** From Tab. 4, *HiPoSeg* improves the baseline by +2.43%p mIoU and achieves an average margin of +4.41%p over other approaches. Compared to HSSN and LogicSeg, the method ensures improvements of +1.01%p and +0.87%p, respectively, reaching the best accuracy (49.33%). Given the fine-grained, imbalanced nature of this dataset, these results underline the ability of *HiPoSeg* to handle rare and small object parts. By enforcing hierarchical alignment, the method reduces inconsistencies between object-level and part-level predictions, yielding more coherent parsing.

### 4.3 ABLATION ANALYSIS

Tab. 5 shows the effect of each loss term. Using only cross-entropy (73.55%) leaves sibling confusions and weak boundaries. High-level contrast (80.96%) structures the parent space and prunes fine-level candidates, while low-level contrast (81.04%) sharpens boundaries and small objects. Combining both without alignment (79.16%) hurts performance due to gradient interference and high–low mismatch. With the proposed alignment (84.04%), the hierarchy is preserved and learning proceeds from coarse to fine: parents stabilize first, then fine classes converge.

Tab. 6 evaluates hierarchical prototypes. Using only parents or only children improves the baseline by regularizing global semantics or refining local details, respectively. Using both yields the best mIoU across Cityscapes, ADE20K, Mapillary, and PASCAL-Part-108 (84.04%, 48.99%, 41.42%, and 49.33%), confirming their complementarity: parent prototypes constrain the candidate space, while child prototypes drive fine-grained separation.

Fig. 3 reports the results of Cityscapes comparing the baseline and *HiPoSeg* on $\text{mIoU}^1$ (low-level labels) and $\text{mIoU}^2$ (high-level labels). *HiPoSeg* exceeds competing methods in both metrics and exhibits a smaller gap between them (9.77%p) than other approaches, suggesting that hierarchical prototype learning mitigates cross-level inconsistencies. Tab. 7 and 8 visualize per-class IoU at high and low levels, respectively; *HiPoSeg* consistently exceeds the baseline in both, indicating strong predictive accuracy throughout the hierarchy. All baseline models are DeepLabV3+.

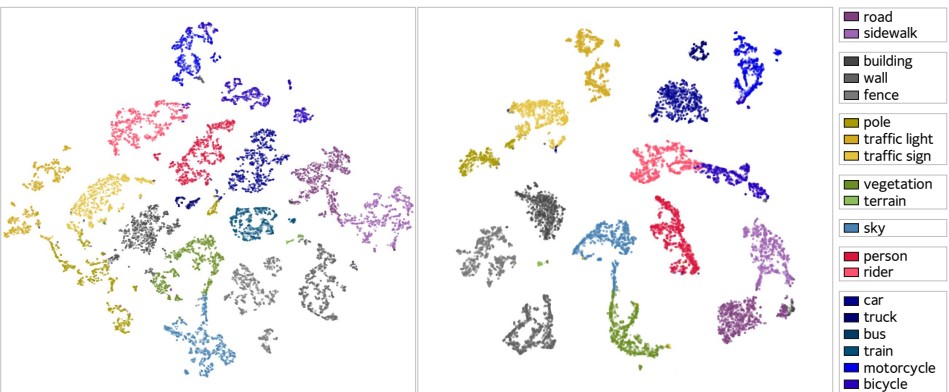

Figure 4: Baseline (left) and our method (right) t-SNE visualization of the feature space on the Cityscapes `val` set. Colors for each class are indicated in the legend on the right, and similar color tones denote classes belonging to the same higher-level concept (e.g., 'road' and 'sidewalk' fall under the higher-level category 'flat').

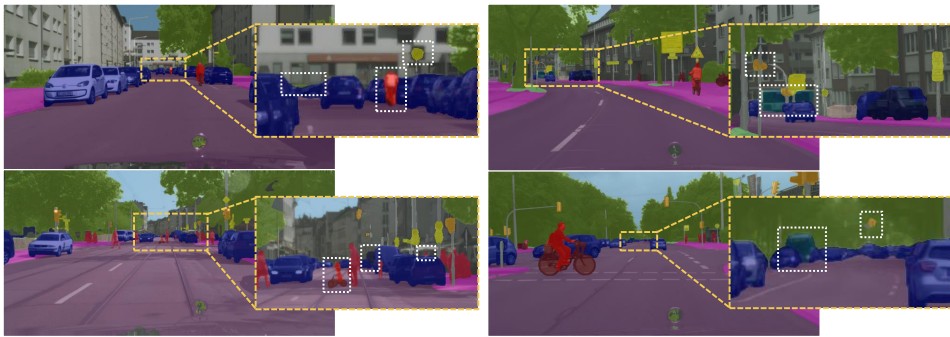

Figure 5: Qualitative results on dense small objects and severe occlusion. Our method clearly segments a bus occluded by a fence or cars, and accurately predicts a traffic light located among vegetation.

## 4.4 QUALITATIVE EVALUATION

Fig. 4 visualizes the feature space on the Cityscapes `val` set using t-SNE for the baseline (left) and *HiPoSeg* (right). Compared to the baseline, which shows heavier overlap among categories and less distinct separation even between semantically related classes, *HiPoSeg* produces a more structured embedding space: classes belonging to the same higher-level concept form more coherent clusters, while different higher-level groups become more clearly separated. This qualitatively supports that *HiPoSeg* injects hierarchy into the representation space itself, learning a clearer coarse-to-fine semantic organization rather than relying on post-hoc correction at the output stage.

In addition, Fig. 5 provides qualitative results in challenging cases with dense small objects and severe occlusion. For instance, our method cleanly segments a bus heavily occluded by a fence or cars, and accurately predicts a small traffic light located among vegetation. These examples suggest that *HiPoSeg* improves not only boundary delineation but also small-object recognition and robustness under severe occlusion.

Fig.6 presents qualitative comparisons between the baseline and the proposed *HiPoSeg* on three challenging benchmarks. *HiPoSeg* delivers cleaner segmentations than the baseline on all datasets, consistently improving boundary sharpness, small-object recognition, and semantic consistency. For instance, in the left Cityscapes example, the baseline mixes the bus with 'building', whereas *HiPoSeg* correctly segments it as 'bus', indicating that hierarchical label learning effectively reduces confusion. Similarly, in the right ADE20K example, the baseline confuses the sea with the sky, while *HiPoSeg* correctly identifies the sea, demonstrating consistent recognition across parent–child cate-

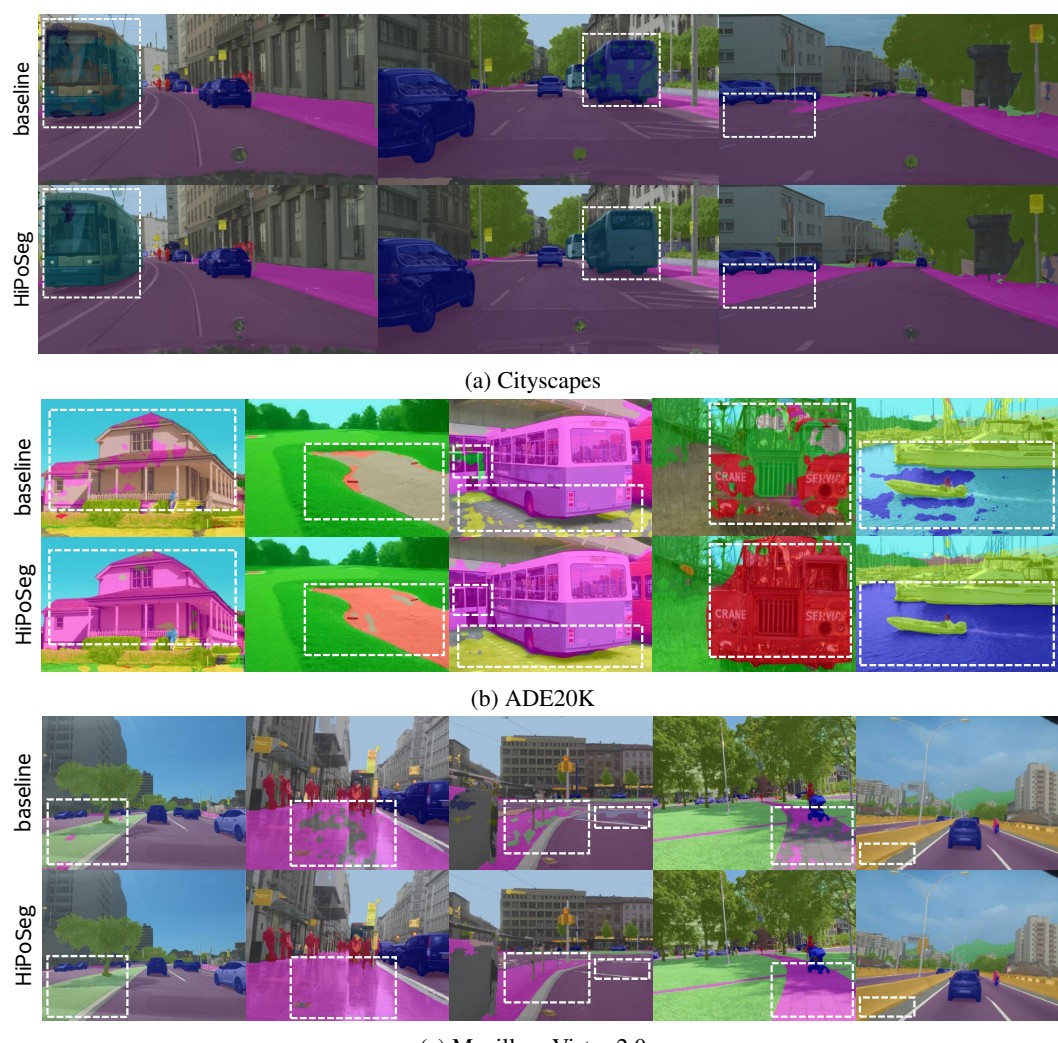

(a) Cityscapes

(b) ADE20K

(c) Mapillary Vistas 2.0

Figure 6: Qualitative case studies on three benchmarks: (a) Cityscapes, (b) ADE20K, and (c) Mapillary Vistas 2.0.

gories. Overall, these qualitative results show that *HiPoSeg* reliably enhances boundary delineation, small object recognition, and semantic consistency across diverse datasets, validating the effectiveness of hierarchical prototype learning for fine-grained segmentation.

## 5 CONCLUDING REMARKS

We have proposed a prototype-based hierarchical method of *HiPoSeg* for semantic segmentation. Unlike flat classification approaches, It organizes class representations into a hierarchical prototype space and enforces cross-level consistency through contrastive learning, enabling a coarse-to-fine recognition process. Extensive experiments on semantic segmentation benchmarks demonstrate that it consistently outperforms strong baselines and previous hierarchy-aware or contrastive methods, with an average gain of +3.07%p mIoU, without incurring additional inference cost. Beyond accuracy, it improves semantic consistency across label hierarchies, reducing confusion between unrelated classes, and enhancing fine-grained part segmentation.

In future work, we will extend *HiPoSeg* to open-vocabulary and long-tailed segmentation, where structured prototypes can mitigate data imbalance, and to multi-modal perception tasks such as vision–language segmentation.

## ACKNOWLEDGEMENTS

This work was supported by the Yonsei Fellow Program funded by Lee Youn Jae, and IITP grant funded by the Korea government (MSIT) (No. RS-2020-II201361, Artificial Intelligence Graduate School Program (Yonsei University); No. RS-2022-II220113, Developing a Sustainable Collaborative Multi-modal Lifelong Learning Framework).

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

APPENDIX

## A  USE OF LLMS

In this work, we used an LLM (ChatGPT) solely as a general-purpose assistive tool. Specifically, it was used to polish the writing, rephrase sentences for more natural and clearer expression, and check grammatical consistency. In addition, it was used to generate the Human and Model illustrations in Fig. 1. Beyond these uses, the LLM was not involved in research ideation, the design of the proposed method, the execution of experiments, or the analysis of results; all such technical contributions were conceived and carried out entirely by the authors.

## B  EXTENDED EXPERIMENTAL RESULTS

**Backbone generalization (Table 9).**  Table 9 summarizes the results on multiple backbones and benchmarks. Our method yields consistent improvements on both HRNet-W48 and SegFormer across the evaluated datasets, supporting backbone-irrelevant behavior. Importantly, we do not introduce any backbone-specific hyperparameters for HRNet-W48 or SegFormer, and we observe no performance degradation in any evaluated configuration. Notably, the gains remain present even on a strong modern architecture such as SegFormer, suggesting that our hierarchical prototype space complements feature formation rather than relying on backbone- or decoder-specific output-stage heuristics. Overall, these results strengthen our claim that the proposed method is a safe, architecture-agnostic plug-in module whose advantages persist across both CNN- and Transformer-based segmentation frameworks.

**Effect of $\sigma_1$ and $\sigma_2$ (Table 10).**  $\sigma_1$ controls the low–high semantic proximity, while $\sigma_2$ regulates the separation among high-level prototypes. As shown in Table 10, inadequate proximity (too small $\sigma_1$) or insufficient separation (too small $\sigma_2$) leads to noticeable performance drops. Overall, $\sigma_1=0.25$ and $\sigma_2=1.00$ provide the best balance between semantic compactness and discrimination, yielding the highest mIoU.

**Momentum coefficient $m$ (Table 11).**  We evaluate how the momentum used in the prototype-memory update affects segmentation performance. As reported in Table 11, a moderate momentum ($m=0.90$) achieves the best trade-off between stability and adaptability, resulting in the highest mIoU among the tested values.

**Staged vs. joint training (Table 12).**  We compare sequential hierarchical learning with a joint optimization setup in Table 12. Activating hierarchical constraints after an initial feature stabilization phase mitigates early prototype drift and leads to consistently better convergence, which translates into higher mIoU. These results confirm that the chosen hyperparameters and the staged activation mechanism are important for stable hierarchical prototype learning and for achieving performance improvements.

**Hierarchical IoU analysis on Mapillary Vistas 2.0.**  We provide a hierarchical breakdown of the segmentation performance on Mapillary Vistas 2.0 `val`. As shown in Table 13 and Table 14, *HiPoSeg* substantially improves the recognition of high-level semantic groups, yielding large gains on several major categories (e.g., *construction*, *object*, and *marking* at the top level, and *vehicle*, *flat*, and *marking_discrete* at the high level). These results indicate that the proposed hierarchical prototype space effectively mitigates broad semantic confusions that are common in large-scale street-scene taxonomies.

**Fine-grained improvements at the low level.**  Table 15 further reports the class-wise IoU at the low level. Beyond the gains at higher abstraction levels, *HiPoSeg* also improves numerous fine-grained classes, including several structure- and marking-related categories as well as small or thin objects (e.g., various traffic-related classes). While a small number of categories show marginal regressions, the overall trend remains strongly positive, suggesting that the hierarchical constraints primarily enhance discrimination without sacrificing fine-detail segmentation.

Table 9: Backbone generalization results across datasets.

| Backbone | Cityscapes | PASCAL-Part-108 | ADE20K | Mapillary |
|---|---|---|---|---|
| ResNet-101 | 73.55 | 46.90 | 44.48 | 31.65 |
| **+ HiPoSeg** | **84.04** | **49.33** | **48.99** | **41.42** |
| HRNet-W48 | 80.85 | 43.77 | 38.59 | 30.17 |
| **+ HiPoSeg** | **81.39** | **43.98** | **38.95** | **43.48** |
| SegFormer | 80.11 | 51.32 | 43.69 | 44.29 |
| **+ HiPoSeg** | **80.63** | **51.59** | **44.29** | **45.04** |

Table 10: Effect of $\sigma_1$ and $\sigma_2$.

| $\sigma_1$ | $\sigma_2$ | mIoU (%) |
|---|---|---|
| 0.10 | 0.50 | 82.30 |
| 0.10 | 1.00 | 82.37 |
| 0.25 | 0.50 | 81.53 |
| 0.25 | 1.00 | **84.04** |

Table 11: Momentum coefficient $m$.

| $m$ | mIoU (%) |
|---|---|
| 0.80 | 81.34 |
| 0.90 | **84.04** |
| 0.99 | 83.28 |

Table 12: Staged vs. joint training.

| training | mIoU (%) |
|---|---|
| joint | 82.47 |
| sequential | **84.04** |

**Qualitative results.** Figure 7 provides qualitative comparisons on Cityscapes, ADE20K, and Mapillary Vistas 2.0. Compared to the baseline, *HiPoSeg* produces clearer object boundaries and more coherent regions, improves segmentation of small or visually ambiguous objects, and reduces confusions among visually similar classes. These observations are consistent with the quantitative improvements reported in Table 13–Table 15, supporting that our method improves both semantic grouping (high-level correctness) and fine-grained delineation (low-level accuracy).

| top-level | baseline (%) | HiPoSeg (%) | gap (%p) |
|---|---|---|---|
| construction | 40.14 | **91.16** | +51.02 |
| human | 36.30 | **71.92** | +35.62 |
| marking | 20.12 | **69.02** | +48.90 |
| nature | 63.50 | **96.06** | +32.56 |
| object | 30.46 | **78.02** | +47.56 |
| void | 40.50 | **62.19** | +21.69 |

Table 13: Top-level IoU comparison on Mapillary Vistas 2.0 `val`.

| high-level | baseline (%) | HiPoSeg (%) | gap (%p) |
|---|---|---|---|
| barrier | 34.07 | **65.98** | +31.91 |
| flat | 39.39 | **89.74** | +50.35 |
| structure | 57.95 | **87.01** | +29.06 |
| person | 40.11 | **71.30** | +31.19 |
| rider | 33.76 | **54.85** | +21.09 |
| marking_continuous | 36.46 | **57.94** | +21.48 |
| marking_discrete | 21.97 | **68.01** | +46.04 |
| nature | 63.50 | **96.06** | +32.56 |
| object | 21.54 | **36.38** | +14.84 |
| sign | 14.70 | **49.39** | +34.69 |
| support | 45.96 | **52.59** | +6.63 |
| traffic_light | 27.64 | **67.07** | +39.43 |
| traffic_sign | 33.61 | **67.04** | +33.43 |
| vehicle | 38.58 | **90.46** | +51.88 |
| void | 40.50 | **62.19** | +21.69 |

Table 14: High-level IoU comparison on Mapillary Vistas 2.0 `val`.

| low-level | baseline (%) | HiPoSeg (%) | gap (%p) | low-level | baseline (%) | HiPoSeg (%) | gap (%p) |
|---|---|---|---|---|---|---|---|
| Concrete Block | 69.79 | **70.32** | +0.53 | Fence | 61.77 | **62.28** | +0.51 |
| Curb | 58.80 | **61.09** | +2.29 | Other Barrier | 0.00 | 0.00 | 0.00 |
| Guard Rail | 63.70 | **64.85** | +1.15 | Road Side | **9.61** | 6.36 | -3.25 |
| Road Median | 14.28 | **14.68** | +0.40 | Temporary Barrier | 38.38 | **41.65** | +3.27 |
| Lane Separator | 17.52 | **19.44** | +1.92 | Bike Lane | 39.64 | **41.52** | +1.88 |
| Wall | 50.21 | **51.28** | +1.07 | Curb Cut | 20.64 | **22.18** | +1.54 |
| Crosswalk - Plain | 31.94 | **33.34** | +1.40 | Parking | **19.24** | 15.61 | -3.63 |
| Driveway | 19.12 | **20.13** | +1.01 | Pedestrian Area | 29.50 | **49.77** | +20.27 |
| Parking Aisle | 0.00 | **3.44** | +3.44 | Road | 87.50 | **87.84** | +0.34 |
| Rail Track | 53.63 | **54.12** | +0.49 | Service Lane | **47.32** | 45.58 | -1.74 |
| Sidewalk | 65.29 | **69.18** | +3.89 | Traffic Island | **38.54** | 36.72 | -1.82 |
| Bridge | 66.28 | **78.33** | +12.05 | Building | 86.63 | **87.84** | +1.21 |
| Tunnel | 12.99 | **65.62** | +52.63 | Person Group | 7.21 | **8.60** | +1.39 |
| Individual | 68.72 | **71.62** | +2.90 | Motorcyclist | 46.21 | **53.89** | +7.68 |
| Bicyclist | 46.52 | **47.39** | +0.87 | Lane Marking - Arrow (Left) | **0.25** | 0.00 | -0.25 |
| Lane Marking - Dashed Line | 43.01 | **46.76** | +3.75 | Lane Marking - Crosswalk | 71.74 | **74.11** | +2.37 |
| Lane Marking - Straight Line | 60.18 | **62.62** | +2.44 | Lane Marking - Hatched (Diagonal) | 48.84 | **50.86** | +2.02 |
| Lane Marking - Arrow (Straight) | **24.89** | 24.74 | -0.15 | Lane Marking - Stop Line | 45.65 | **50.05** | +4.40 |
| Lane Marking - Hatched (Chevron) | **37.13** | 35.83 | -1.30 | Lane Marking - Symbol (Bicycle) | 44.88 | **49.79** | +4.91 |
| Lane Marking - Other | 34.92 | **38.29** | +3.37 | Lane Marking - Text | 41.96 | **49.74** | +7.78 |
| Mountain | 45.05 | **50.99** | +5.94 | Sky | 97.85 | **97.86** | +0.01 |
| Sand | 0.00 | **0.01** | +0.01 | Terrain | **65.28** | 65.06 | -0.22 |
| Snow | 73.35 | **75.98** | +2.63 | Vegetation | 89.37 | **89.92** | +0.55 |
| Water | **78.67** | 64.65 | -14.02 | Banner | 28.66 | **31.56** | +2.90 |
| Bench | **39.21** | 38.42 | -0.79 | Bike Rack | 0.00 | **0.52** | +0.52 |
| Fire Hydrant | 9.28 | **51.28** | +42.00 | Catch Basin | **39.96** | 39.78 | -0.18 |
| Junction Box | 36.77 | **44.92** | +8.15 | Manhole | 51.55 | **51.97** | +0.42 |
| Signage - Advertisement | 45.34 | **47.38** | +2.04 | Signage - Back | 0.02 | **0.10** | +0.08 |
| Signage - Information | **0.52** | 1.68 | +1.16 | Signage - Store | **39.59** | 39.05 | -0.54 |
| Street Light | 47.31 | **48.53** | +1.22 | Pole | 42.06 | **46.85** | +4.79 |
| Pole Group | 9.10 | **10.74** | +1.64 | Traffic Sign Frame | 51.47 | **56.43** | +4.96 |
| Utility Pole | 45.65 | **51.58** | +5.93 | Traffic Cone | 56.61 | **61.60** | +4.99 |
| Traffic Light - Pedestrians | 40.36 | **47.35** | +6.99 | Traffic Light - General (Horizontal) | 47.63 | **50.71** | +3.08 |
| Traffic Light - General (Upright) | 64.12 | **67.79** | +3.67 | Traffic Light - Direction (Back) | 41.74 | **44.38** | +2.64 |
| Traffic Sign - Ambiguous | 0.00 | **1.82** | +1.82 | Traffic Sign - Direction (Front) | 69.94 | **71.07** | +1.13 |
| Traffic Sign (Front) | 59.28 | **62.52** | +3.24 | Traffic Sign - Parking | 17.55 | **34.20** | +16.65 |
| Traffic Sign - Temporary (Front) | 0.00 | **10.72** | +10.72 | Trash Can | 41.74 | **42.68** | +0.94 |
| Bicycle | 53.57 | **58.20** | +4.63 | Boat | 1.71 | **20.46** | +18.75 |
| Bus | 75.05 | **79.55** | +4.50 | Caravan | 0.00 | 0.00 | 0.00 |
| Motorcycle | 53.16 | **57.91** | +4.75 | On Rails | 28.84 | **46.50** | +17.66 |
| Other Vehicle | 17.13 | **18.22** | +1.09 | Truck | 71.55 | **73.96** | +2.41 |
| Vehicle Group | 16.63 | **17.50** | +0.87 | Dynamic | **29.61** | 29.11 | -0.50 |
| Ego Vehicle | 89.18 | **92.33** | +3.15 | Ground | 30.56 | **33.29** | +2.73 |
| Static | 18.74 | **20.23** | +1.49 | Unlabeled | 0.37 | **0.55** | +0.18 |

Table 15: Low-level IoU comparison between baseline and *HiPoSeg* on the Mapillary Vistas 2.0 `val` set.

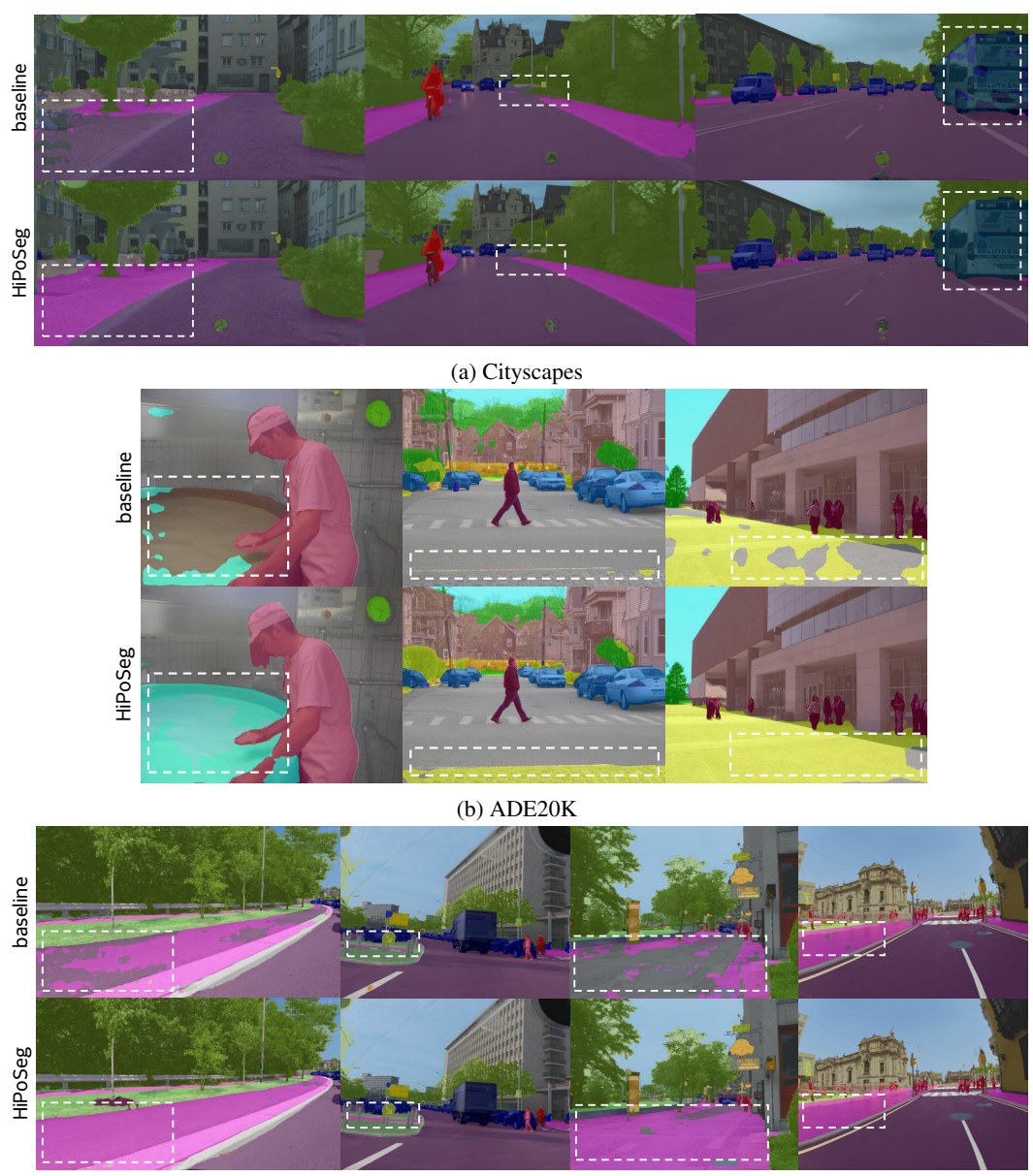

(a) Cityscapes

(b) ADE20K

(c) Mapillary Vistas 2.0

Figure 7: Compared to the baseline, *HiPoSeg* yields clearer boundaries, improved small-object segmentation, and fewer confusions among visually similar classes.

