# OpenReview forum: "Hierarchical Prototype Learning for Semantic Segmentation"
_ICLR.cc/2026/Conference — ICLR 2026 Poster_

### Official Review · Reviewer_LYNi · 2025-10-30

**Soundness:** 3
**Presentation:** 2
**Contribution:** 2
**Rating:** 4
**Confidence:** 2

**Summary:**

Traditional semantic segmentation methods often overlook semantic relationships across levels. To address this issue, the article proposes a Hierarchical Prototype Learning (HiPoSeg) method. By introducing high- and low-level prototype contrastive learning and high- and low-level alignment constraints, it effectively combines coarse categories and fine-grained part semantics, improving segmentation accuracy and consistency. Experiments across multiple datasets validate the proposed HiPoSeg method's ability for fine-grained segmentation and semantic consistency.

**Strengths:**

The paper effectively integrates coarse- and fine-grained semantics via hierarchical contrastive learning, thereby improving segmentation accuracy.

It enhances semantic consistency across different levels by employing alignment constraints between high- and low-level features.

**Weaknesses:**

- In Section 3.2, regarding the setting of some hyperparameters (e.g., "σ1=0.25, σ2=1" and "m=0.9"), providing an analysis of how the choice and adjustment of these hyperparameters affect performance would help enhance the reproducibility and understanding of the method. Additionally, the paper introduces a step-by-step activation mechanism at different stages (from feature representation to prototype learning, and then to alignment loss). It is recommended to provide more experimental or theoretical support to clarify why this staged training arrangement effectively facilitates hierarchical learning.

- The explanation in Section 3 could benefit from more step-by-step details, especially regarding prototype construction. Although Figure 2 provides a good overview of the overall structure, adding more detailed annotations and explanations would help readers better understand each step.

- The latest relevant methods are missing from the comparison.

**Questions:**

- Some table captions are above the tables, while others are below. It is recommended to review and adjust the table layout for greater clarity.

- In the first paragraph of Section 3, "...prototype space in 3.1. Second, in 3.2..." should be revised to "...prototype space in Sec. 3.1. Second, in Sec. 3.2...“.

- There is an unnumbered equation between Equation 1 and Equation 2.

- Table 7 should be a figure, not a table.

---

> ### Author Response · Authors · 2025-12-02
>
> Thank you for your constructive review. We greatly appreciate that you accurately recognized HiPoSeg’s key idea of *effectively combining coarse categories and fine-grained part semantics through high-/low-level prototype-based contrastive learning and cross-level alignment constraints.*
>
> ---
> ***W1. Hyperparameter Sensitivity and Training Schedule Analysis***
>
> We have added detailed analyses of hyperparameters and training schedule to improve reproducibility and clarify the rationale behind our staged design.
>
> **1. Effect of σ₁ and σ₂**
>
> σ₁ controls low–high semantic proximity, while σ₂ regulates separation among high-level prototypes. As shown below, insufficient proximity or insufficient separation leads to performance drops. Thus, σ₁=0.25 and σ₂=1.00 provide the best balance between semantic compactness and discrimination.
>
> |  σ₁  |  σ₂  | mIoU (%) |
> |:---:|:---:|:---:|
> | 0.10 | 0.50 | 82.30  |
> | 0.10 | 1.00 | 82.37 |
> | 0.25 | 0.50 | 81.53 |
> | **0.25** | **1.00** | **84.04** |
>
> **2. Momentum coefficient m**
>
> We have evaluated how the momentum in prototype memory update affects performance. A moderate momentum (m=0.90) yields the best trade-off between stability and adaptability.
>
> | m    | mIoU (%) |
> |:----:|:---------:|
> | 0.80 | 81.34    |
> | **0.90** | **84.04**    |
> | 0.99 | 83.28    |
>
> **3. Staged vs. joint training**
>
> We have compared sequential hierarchical learning with a joint optimization setup. Activating hierarchical constraints after initial feature stabilization prevents early prototype drift and results in consistently better convergence.
>
> | threshold            | mIoU (%) |
> |:---------------------|:---------:|
> | joint                | 82.47    |
> | **sequential learning**  | **84.04**    |
>
> These results confirm that the selected hyperparameters and staged activation mechanism are essential for stable hierarchical prototype learning and performance improvement.
>
> ---
> ***W2. Step-by-step Detailed Explanation of Prototype Learning***
>
> We appreciate the comment regarding procedural clarity. While `Sec. 3` already describes the overall workflow, we add a clearer step-by-step summary by organizing the prototype learning process into four stages:
>
> **1. Prototype construction:** Pixel features are grouped by low-level labels and their corresponding high-level mappings to initialize both low- and high-level prototypes.
>
> **2. High-level learning:** Pixel features are pulled toward their high-level prototypes that are enforced to be mutually separated.
>
> **3. Low-level learning:** After high-level convergence, low-level discrimination is refined using the same pull–push mechanism.
>
> **4. Hierarchical alignment:** Each low-level prototype is constrained to remain close to its parent high-level prototype while preserving separation between different high-level groups.
>
> For better readability, we have also updated the caption in `Figure 2` to make these stages easier to follow at a glance.
>
> ---
> ***W3. Comparison with the Latest Related Work***
>
> We would like to argue that the comparison is **not missing the latest relevant methods** within the scope of this paper, namely fully supervised semantic segmentation under fixed label sets. Our experimental results already include recent and competitive baselines in this setting, including Contextrast [1], along with other strong recent supervised segmentation methods.
>
> We have briefly stressed the key distinction that motivates our method relative to these recent baselines. Many recent contrastive/representation-learning approaches improve segmentation primarily through context- or sampling-centric designs that strengthen pixel/region representations, typically assuming a flat label space. In contrast, HiPoSeg is explicitly designed to leverage hierarchical supervision by maintaining high- and low-level prototype memories and introducing a hierarchy-aware alignment mechanism that structures the embedding space in a coarse-to-fine manner during training. We believe this difference in supervision target and representation structuring explains why HiPoSeg achieves consistent improvements against the most recent supervised baselines in our comparison.
>
> ---
> ***Q1~4. Paper Notation and Layout Issues***
>
> We appreciate the reviewer’s careful comments. All four issues have been addressed in the revised paper as follows:
>
> (Q1) All table captions have been unified to appear below the tables.
>
> (Q2) The sentence in `Sec. 3` has been corrected to “prototype space in Sec. 3.1 … in Sec. 3.2”.
>
> (Q3) The previously unnumbered equation between `Eq. 1` and `Eq. 2` is now properly numbered.
>
> (Q4) `Table 7` has been reformatted as a figure (`Fig. 3`).
>
> ---
> [1] Sung, Changki, et al. "Contextrast: Contextual contrastive learning for semantic segmentation." Proceedings of the IEEE/CVF conference on computer vision and pattern recognition. 2024.

---

### Official Review · Reviewer_Bxs7 · 2025-10-30

**Soundness:** 3
**Presentation:** 3
**Contribution:** 2
**Rating:** 4
**Confidence:** 2

**Summary:**

* The paper introduces Hierarchical Prototype Segmentation (HiPoSeg), a new framework for semantic segmentation that models visual recognition as a coarse-to-fine process, reflecting how humans perceive objects.
* HiPoSeg builds a hierarchical prototype space that captures both high-level object semantics and fine-grained part semantics, linking them through a hierarchical contrastive learning objective. To maintain consistency across these levels, the model applies a multi-level alignment constraint, reducing misclassifications between unrelated categories.
* A key strength of HiPoSeg is that it functions as a training-only, plug-and-play module, meaning it adds no extra computational cost during inference while still enhancing semantic consistency and mean Intersection over Union (mIoU).
* Tested on four major benchmarks, Cityscapes, ADE20K, Mapillary Vistas 2.0, and PASCAL-Part-108, HiPoSeg consistently outperforms strong baselines such as HSSN, LogicSeg, and ProtoSeg, achieving notable improvements across all datasets.

**Strengths:**

* The paper clearly formalizes hierarchical semantics by organizing prototypes in a top-down hierarchy, addressing the structural limitations of flat segmentation.
* The proposed component is used only during training, requiring no additional computation or parameters at inference, which is practically appealing.
* Cross-level semantic consistency: Through an alignment constraint, HiPoSeg maintains strong coherence between high-level and low-level features, minimizing semantic drift and enhancing the precision of fine-grained segmentation.
* Comprehensive validation: The paper provides extensive quantitative and qualitative analyses across multiple benchmark datasets, clearly demonstrating the robustness and generality of the method.
* HiPoSeg achieves higher mIoU than SoTA models (HSSN, LogicSeg, ProtoSeg, ContextSeg), even with a weaker backbone (ResNet-101), underscoring its efficiency and scalability.

**Weaknesses:**

* While the hierarchical prototype framework is well-motivated, it conceptually overlaps with prior hierarchical reasoning models such as HSSN (CVPR 2022) and LogicSeg (ICCV 2023). The improvement may be seen as incremental rather than fundamentally new.
* The experiments focus primarily on two-level hierarchies. The extension to deeper hierarchies (e.g., three-level settings in ADE20K and Mapillary) is only lightly discussed, leaving generalization uncertain.
* Although the paper presents loss-wise ablations, it lacks an analysis of hyperparameter sensitivity.
* The qualitative comparisons rely heavily on visual inspection of boundary sharpness and small-object recognition, which are somewhat subjective and lack quantitative support.
* Stability under multi-term losses: The hierarchical contrastive loss combines multiple objectives (Lfh, Lfl, Lalign), which may cause gradient interference in large-scale training unless carefully balanced.
* HiPoSeg assumes a fixed and known label tree, making it less applicable to datasets without clear hierarchical semantics or dynamically evolving taxonomies.
* Although mentioned in the conclusion, the paper does not empirically validate HiPoSeg in open-vocabulary or long-tailed scenarios.

**Questions:**

Can the hierarchical prototype space be learned when the label hierarchy is not predefined?
What is the rationale behind the curriculum schedule for enabling hierarchical components, and how sensitive is the model to these thresholds?

---

> ### Author Response · Authors · 2025-12-02
>
> We sincerely appreciate the reviewer’s thoughtful comments on our central goal of *connecting coarse object categories with fine-grained part semantics via multi-level prototype learning and cross-level alignment.*
>
> ---
> ***W1. Comparison with Existing Methods Using Hierarchical Structures***
>
> Although HSSN [1] and LogicSeg [2] also incorporate hierarchical information, our method differs fundamentally in that the hierarchy is injected in the segmentation pipeline. HSSN [1] and LogicSeg [2] apply hierarchy after feature extraction—either at the logit level or through a symbolic reasoning module—so the representation space itself is not explicitly reorganized during training.
>
> Our method instead introduces **hierarchy at the representation formation stage by learning a hierarchical prototype space.** Subclasses within the same coarse category are encouraged to cluster together while remaining separated from other coarse groups, enabling hierarchical structure to emerge directly in the embedding manifold rather than being imposed post-hoc on output logits.
>
> This representation-level restructuring leads to clear practical benefits: reduced feature overlap within coarse groups and more robust feature geometry for rare subclasses, even **without inference-time reasoning.** Empirically, this manifests as significant gains in fine-grained segmentation accuracy, **+10.49%** mIoU on Cityscapes and **+4.51%** mIoU on ADE20K—with particularly large improvements on rare categories such as train and truck.
>
> Prototype–feature distance distributions and t-SNE plots in the main paper further confirm that hierarchical separation is consistently formed in the learned feature space, a behavior not observed in HSSN [1] or LogicSeg [2].
>
> ---
> ***W2. Additional Clarification on the Two-level/Three-level Setting in the Dataset***
>
> As described in `Sec. 4.1`, Cityscapes and PASCAL-Part-108 are naturally organized with two label levels, whereas ADE20K and Mapillary Vistas 2.0 provide three label levels. However, since our method only requires upper-level semantics for the low-level classes, we consistently adopt a two-level label setting across all four datasets. For ADE20K and Mapillary Vistas 2.0, we define the high-level labels using the mid-level and fine-grained classes, respectively, to construct the required parent–child relations with the low-level labels. We have added this clarification and the corresponding dataset setup details to `Sec. 4.1`.
>
> ---
> ***W3. (Q1) Hyperparameter Sensitivity Analysis***
>
> We have added detailed analyses of hyperparameters to improve reproducibility and clarify the rationale behind our staged design.
>
> **1. Effect of σ₁ and σ₂**
>
> σ₁ controls low–high semantic proximity, while σ₂ regulates separation among high-level prototypes. As shown below, insufficient proximity or insufficient separation leads to performance drops. Thus, σ₁=0.25 and σ₂=1.00 provide the best balance between semantic compactness and discrimination.
>
> |  σ₁  |  σ₂  | mIoU (%) |
> |:---:|:---:|:---:|
> | 0.10 | 0.50 | 82.30  |
> | 0.10 | 1.00 | 82.37 |
> | 0.25 | 0.50 | 81.53 |
> | **0.25** | **1.00** | **84.04** |
>
> **2. Momentum coefficient m**
>
> We have evaluated how the momentum in prototype memory update affects performance. A moderate momentum (m=0.90) yields the best trade-off between stability and adaptability.
>
> | m    | mIoU (%) |
> |:----:|:---------:|
> | 0.80 | 81.34    |
> | **0.90** | **84.04**    |
> | 0.99 | 83.28    |
>
> These results confirm that the selected hyperparameters are essential for stable hierarchical prototype learning and performance improvement.
>
> ---
> ***W4. Quantitative Evidence to Support the Qualitative Results***
>
> Thank you for the comment. To reduce the subjectivity of qualitative visual inspection (e.g., boundary sharpness and small-object recognition), we have added quantitative evidence via a supplementary table that reports low-level, fine-grained class-wise IoU on the Mapillary Vistas 2.0 val set. This provides direct, per-category measurements for many small or thin-structure classes where qualitative differences are most noticeable.
>
> As shown in `Table 9`, HiPoSeg improves IoU on numerous fine-grained categories, including Tunnel (**+52.63%p**), Fire Hydrant (**+42.00%p**), Pedestrian Area (**+20.27%p**), Bridge (**+12.05%p**), Traffic Sign-Parking (**+16.65%p**), Boat (**+18.75%p**), and On Rails (**+17.66%p**), which quantitatively supports the qualitative analyses.
>
> ---
> [1] Li, Liulei, et al. "Deep hierarchical semantic segmentation." Proceedings of the IEEE/CVF conference on computer vision and pattern recognition. 2022.
>
> [2] Li, Liulei, Wenguan Wang, and Yi Yang. "Logicseg: Parsing visual semantics with neural logic learning and reasoning." Proceedings of the IEEE/CVF international conference on computer vision. 2023.

---

> > ### Author Response · Authors · 2025-12-02
> >
> > ***W5. Explanation of Using Multiple Loss Terms***
> >
> > Although the overall objective may appear to include many loss terms, **these terms are not an arbitrary accumulation of independent goals.** Rather, they explicitly decompose our training objective into four components: (i) standard segmentation learning (CE), (ii) high-level prototype learning, (iii) low-level prototype learning, and (iv) hierarchical alignment. Concretely, $L_{fh}$ and $L_{hh}$ implement, respectively, *feature-to-prototype alignment* and *prototype separation* for high-level prototype learning, while $L_{fl}$ and $L_{ll}$ play the same roles for low-level prototype learning. Thus, although the objective contains multiple terms, it is essentially a symmetric application of the same (alignment/separation) structure at both the high and low semantic levels.
> >
> > In addition, our method mitigates potential conflicts among these terms by adopting the stage-wise training schedule described in `Sec. 3.2 (lines 277–282)`. We first stabilize the backbone/decoder representations using the CE loss, then activate high-level prototype learning, and subsequently introduce low-level prototype learning and alignment sequentially. This prevents a scenario where all terms simultaneously produce large gradients from the beginning of training.
> >
> > ---
> > ***W6. Dependence on a Fixed Label Tree***
> >
> > As you pointed out previously, HiPoSeg leverages a predefined hierarchy (i.e., a given label tree). However, **the goal of this work is to reduce coarse–fine inconsistency and semantic leakage caused by flat classification under a supervised semantic segmentation setting** where hierarchical semantics are explicitly provided (e.g., object–part relations, parent–child taxonomies). Therefore, scenarios in which the hierarchy is unclear or dynamically evolving fall outside the problem setting assumed in this paper (e.g., open-vocabulary segmentation, incremental taxonomies, and hierarchy discovery), and we do not claim that our method directly applies to those settings.
> >
> > We also explicitly mention such cases—where hierarchies are ambiguous or dynamically changing—as future work in `Sec. 5 (lines 538–539)`.
> >
> > ---
> > ***W7. Validation in Open-Vocabulary and Long-Tailed Settings***
> >
> > Thank you for the comment. **Open-vocabulary and long-tailed segmentation are fundamentally different from the supervised closed-set setting considered** in this paper in both problem definition and evaluation protocol. Therefore, they are not a suitable setting to directly validate our core claim—improving coarse–fine consistency through hierarchical prototype learning—on the same footing as our current experiments.
> >
> > For this reason, we focus on standard benchmarks with a predefined hierarchy, and we explicitly state in `Sec. 5 (lines 538–539)` that extending HiPoSeg to open-vocabulary and long-tailed scenarios is an interesting direction of future work.

---

### Official Review · Reviewer_5rqn · 2025-10-31

**Soundness:** 3
**Presentation:** 3
**Contribution:** 3
**Rating:** 4
**Confidence:** 4

**Summary:**

This paper introduce HiPoSeg, a approach to semantic segmentation that leverages hierarchical prototype learning to improve segmentation performance, particularly for fine-grained tasks. The key idea is to model both high-level categories and their subcategories through hierarchical prototypes, allowing the model to capture both coarse and fine-grained semantic structures.

**Strengths:**

* The paper introduces hierarchical prototype learning for semantic segmentation, combining prototype-based learning with a hierarchical structure, which improves the model’s ability to differentiate fine-grained subcategories.
* The problem is clearly defined, and the paper is well-structured,  presenting experimental results in an understandable manner.
* HiPoSeg improves segmentation results, and its scalability and flexibility suggest broad potential for various applications.

**Weaknesses:**

* This paper uses DeepLabv3, a classic but somewhat outdated baseline. To validate the generalizability of this method, the authors should consider using a wider variety of CNNs and Transformer-based baselines. Also, this can demonstrate the author's claim of "plug-and-play efficiency".

* The paper mentions the novelty of hierarchical prototype learning, but it lacks a discussion with other similar methods, such as other prototype-based semantic segmentation methods or hierarchical network approaches.

* It would be useful to clearly distinguish the fundamental differences between this approach and existing hierarchical methods, for example, does the prototype learning mechanism in HiPoSeg offer better generalization ability? Does it form a clearer semantic structure in the feature space?  Does HiPoSeg more effectively leverage logical constraints or hierarchical consistency?

* It may be better to add visual analysis (e.g., t-SNE or prototype space structure plots) to demonstrate how the prototype space is organized and whether the hierarchical structure is reflected in the learned representations.

* While the paper introduces hierarchical prototype learning for handling fine-grained segmentation tasks, there has not been sufficient exploration of its performance on fine-grained subcategories and its advantages in these tasks.


I would be happy to revise my score if the author addresses these points.

**Questions:**

Please refer to the weakness.

---

> ### Author Response · Authors · 2025-12-02
>
> Thank you for the valuable comments. We appreciate your recognition of our *clear problem formulation*, the effectiveness of *hierarchical prototype learning for fine-grained segmentation, and the strong scalability and applicability of HiPoSeg.*
>
> ---
> ***W1. Backbone Diversity and Plug-and-Play Generalization***
>
> We appreciate the reviewer’s suggestion to assess backbone diversity for validating plug-and-play generalizability. In response, we have additionally evaluated two architecturally different backbone families—HRNet-W48 and SegFormer—and confirmed that HiPoSeg consistently improves performance beyond the original ResNet-based setting.
>
> **1. Quantitative results across datasets**
>
> As summarized below, our method provides consistent improvements on both HRNet-W48 and SegFormer across multiple benchmarks, supporting backbone irrelevancy.
>
> | Backbone    | Cityscapes | PASCAL-Part-108 | ADE20K | Mapillary |
> |:-----------:|:-----------:|:----------------:|:------:|:----------:|
> | ResNet-101 | 73.55      | 46.90           | 44.48 | 31.65     |
> | **+Ours**      | **84.04**      | **49.33**           | **48.99** | **41.42**     |
> | HRNet-W48  | 80.85      | 43.77           | 38.59 | 30.17     |
> | **+Ours**      | **81.39**      | **43.98**           | **38.95** | **43.48**     |
> | SegFormer  | 80.11      | 51.32           | 43.69 | 44.29     |
> | **+Ours**      | **80.63**      | **51.59**           | **44.29** | **45.04**     |
>
> **2. Additional clarifications**
>
> No backbone-specific hyperparameters are introduced for HRNet-W48 or SegFormer. We have observed no performance degradation in any evaluated configuration.
>
> Gains remain present even on a strong modern architecture such as SegFormer, which suggests that our hierarchical prototype space complements feature formation rather than relying on backbone- or decoder-specific output-stage heuristics.
>
> Overall, these results strengthen our original claim that the proposed method is a safe, architecture-agnostic plug-in module, whose advantages persist across both CNN- and Transformer-based segmentation frameworks.
>
> ---
> ***W2. Comparison to Prototype-based and Hierarchical Segmentation Methods***
>
> Although `Sec. 2.2` and `Sec. 2.3` **already provide an overview and comparison of prototype-based methods and hierarchical network approaches**, we further elaborate the differences more concretely, following the reviewer’s request, using representative prior works—ProtoSeg [1], HSSN [2], and LogicSeg [3]—as reference points.
>
> Although HSSN [2] and LogicSeg [3] also incorporate hierarchical information, our method differs fundamentally in that the hierarchy is injected in the segmentation pipeline. HSSN [2] and LogicSeg [3] apply hierarchy after feature extraction—either at the logit level or through a symbolic reasoning module—so the representation space itself is not explicitly reorganized during training.
>
> Our method instead introduces **hierarchy at the representation formation stage by learning a hierarchical prototype space.** Subclasses within the same coarse category are encouraged to cluster together while remaining separated from other coarse groups, enabling hierarchical structure to emerge directly in the embedding manifold rather than being imposed post-hoc on output logits.
>
> This representation-level restructuring leads to clear practical benefits: reduced feature overlap within coarse groups and more robust feature geometry for rare subclasses, even **without inference-time reasoning.** Empirically, this manifests as significant gains in fine-grained segmentation accuracy, **+10.49%** mIoU on Cityscapes and **+4.51%** mIoU on ADE20K—with particularly large improvements on rare categories such as train and truck.
>
> Prototype–feature distance distributions and t-SNE plots in the main paper further confirm that hierarchical separation is consistently formed in the learned feature space, a phenomenon not observed in HSSN [2] or LogicSeg [3].
>
> ---
> [1] Zhou, Tianfei, et al. "Rethinking semantic segmentation: A prototype view." Proceedings of the IEEE/CVF conference on computer vision and pattern recognition. 2022.
>
> [2] Li, Liulei, et al. "Deep hierarchical semantic segmentation." Proceedings of the IEEE/CVF conference on computer vision and pattern recognition. 2022.
>
> [3] Li, Liulei, Wenguan Wang, and Yi Yang. "Logicseg: Parsing visual semantics with neural logic learning and reasoning." Proceedings of the IEEE/CVF international conference on computer vision. 2023.

---

> > ### Author Response · Authors · 2025-12-02
> >
> > ***W3. Clear Differentiation from Prior Hierarchical Approaches***
> >
> > We clarify the *fundamental differences from prior hierarchical methods* more explicitly in the main paper. In many existing hierarchical segmentation approaches, the hierarchy is reflected at the level of output probabilities, post-processing, or rule-based constraints, so that the final predictions are forced to satisfy the tree constraints. In contrast, HiPoSeg does not impose the hierarchy as a post-hoc constraint at inference time. Instead, it explicitly constructs high- and low-level prototypes and structures the class representations themselves via hierarchical contrastive learning.
> >
> > Moreover, we introduce cross-level alignment to inject hierarchical consistency directly into the feature space as a learning signal (suppressing semantic leakage). In other words, the key difference is that the hierarchy is not treated as a rule/constraint, but is internalized into the geometry of the representation space.
> >
> > In addition, to illustrate how this difference manifests in practice, we have added `Figure 4` in `Sec. 4.4` to compare the feature spaces of the baseline and HiPoSeg. Compared to the baseline, **HiPoSeg yields better class-wise feature separation while also grouping classes that share the same high-level semantics closer together**, indicating that the model learns a more semantically structured embedding space.
> >
> > ---
> > ***W4. Prototype Space Visualization and Analysis***
> >
> > We have added a comparative t-SNE visualization analysis between the baseline and our method in `Sec. 4.4` of the main paper.
> >
> > ---
> > ***W5. Fine-grained Subcategory Evaluation***
> >
> > Thank you for highlighting the need for a more thorough exploration of our performance on fine-grained subcategories and the advantages of HiPoSeg in these settings. To address this, we have added a supplementary table reporting class-wise IoU on the Mapillary Vistas 2.0 val set at the low-level granularity (`Table 9`). As shown in the table, HiPoSeg achieves consistent improvements over the baseline across many fine-grained classes, with particularly large gains on visually challenging subcategories and small/rare structures. For example, we observe substantial improvements on *Tunnel* (**+52.63%p**), *Fire Hydrant* (**+42.00%p**), *Pedestrian Area* (**+20.27%p**), *Traffic Sign-Parking* (**+16.65%p**), *Boat* (**+18.75%p**), and *On Rails* (**+17.66%p**).
> >
> > These results support the advantage of HiPoSeg by structuring class representations via hierarchical prototype learning and suppressing semantic leakage through cross-level alignment, it strengthens feature separability among fine-grained subcategories.

---

### Official Review · Reviewer_hVxW · 2025-10-31

**Soundness:** 3
**Presentation:** 3
**Contribution:** 3
**Rating:** 6
**Confidence:** 4

**Summary:**

Existing semantic segmentation methods often treat the task as flat classification, ignoring structural relationships between classes and failing to replicate human “whole-to-part” hierarchical recognition, leading to poor fine-grained part segmentation. To address this, the paper proposes HiPoSeg, a hierarchical prototype-based method that constructs a structured prototype space capturing object-level and part-level features, uses hierarchical contrastive learning for intra-level discrimination and cross-level consistency, and only acts during training (no inference overhead). Experiments on Cityscapes, ADE20K, Mapillary Vistas 2.0, and PASCAL-Part-108 show an average +3.07%p mIoU gain over baselines and state-of-the-art methods. Its core contributions include: 1) hierarchical prototype learning for structured class representation; 2) multi-level alignment constraints to suppress semantic leakage; 3) a plug-and-play, training-only design with zero inference cost.

**Strengths:**

1. Unlike prior hierarchical segmentation methods (e.g., HSSN, LogicSeg) that handle hierarchy as a fixed auxiliary term or fuse signals at the probability level, HiPoSeg explicitly constructs a hierarchical prototype space and uses contrastive learning to structure class representations—this design principle is a meaningful departure from existing work.
2. The experimental design is relatively systematic: it validates performance on 4 diverse benchmarks (urban scenes, daily scenes, part parsing), and ablation experiments (Table 5-6) confirm the necessity of high-level/low-level contrastive losses and alignment constraints, ensuring the method’s core components are evidence-based.
3. The method description is detailed: Section 3 clearly defines prototype construction (Eqs. 1-4), hierarchical contrastive learning (high/low-level losses), and alignment constraints (Eqs. for 𝒱ₐₗᵢ₉ₙ), making the technical implementation reproducible in principle.
4. The “zero inference overhead” design is practically valuable—HiPoSeg acts as an auxiliary training component, avoiding the common trade-off between performance gain and deployment cost in segmentation research, which is useful for real-world applications (e.g., autonomous driving).

**Weaknesses:**

1. Experimental Completeness: No hyperparameter sensitivity analysis: The paper sets σ₁=0.25, σ₂=1, m=0.9 without justifying why these values are optimal or how performance changes if they vary (e.g., σ₁=0.1 vs. 0.5).
Lack of alternative backbone validation: All experiments use ResNet-101, but no results on mainstream backbones like HRNet-W48 or SegFormer are provided, making it unclear if HiPoSeg’s gains generalize beyond ResNet.
2. Insufficient qualitative challenging cases: The qualitative results (Figure 3/4) only show simple scenes (e.g., clear buses in Cityscapes) and omit scenarios with dense small objects (e.g., crowded pedestrians) or severe occlusion—critical for verifying fine-grained segmentation ability.
3. Motivation and Context:The related work section does not contrast HiPoSeg with 2024 SOTA methods (e.g., Contextrast’s contextual contrastive learning) in depth, only listing them in tables. It fails to explain why HiPoSeg’s hierarchical prototype design outperforms these latest methods, weakening the motivation’s urgency.
4. No failure case analysis: The paper only reports successful segmentation results but does not analyze when HiPoSeg fails (e.g., which classes/regions still have high error rates) or why, limiting insights into the method’s limitations.
5. Theoretical Depth:The paper does not provide quantitative analysis of prototype space (e.g., t-SNE visualizations of high/low-level prototypes or feature similarity metrics between related classes like “truck” and “bus”). It only claims “structured representation” without empirical evidence of how prototypes reduce cross-class confusion.

**Questions:**

1. Experimental Design:
Could you provide results of HiPoSeg on alternative backbones (e.g., HRNet-W48, SegFormer-B5) to verify its generalization? If performance drops on certain backbones, what is the root cause?
Could you add a hyperparameter sensitivity study (varying σ₁, σ₂, m, and temperature τ/κ) and explain how you selected the optimal values?
Could you supplement qualitative results on challenging scenes (dense small objects, severe occlusion) and analyze error cases (e.g., which low-level classes still have low IoU)?
2. Motivation and Comparison:
How does HiPoSeg’s hierarchical prototype design specifically outperform 2024 SOTA methods like Contextrast (which uses contextual contrastive learning)? Could you add a head-to-head comparison with these methods on the same benchmarks?
Prior work (e.g., ProtoSeg) also uses prototypes—why is combining prototypes with hierarchy more effective than flat prototype learning? Could you provide a controlled experiment comparing HiPoSeg with a flat prototype baseline?
3. Theoretical and Mechanistic Analysis:
Could you provide t-SNE or UMAP visualizations of the prototype space to show how high/low-level prototypes of related classes (e.g., “car”/“truck” in high-level, “horse head”/“horse leg” in low-level) are structured?
For rare low-level classes (e.g., “train” in Cityscapes, which has a baseline IoU of 23.88%), how does HiPoSeg’s prototype memory bank address data sparsity? Could you analyze the prototype update process for these classes?

---

> ### Author Response · Authors · 2025-12-02
>
> We truly appreciate the reviewer’s careful review and helpful comments. Thank you for precisely recognizing HiPoSeg’s core idea of *learning a coarse-to-fine semantic structure through a hierarchical prototype space, hierarchical contrastive learning, and multi-level alignment constraints.* We also appreciate your comments that *our experimental design is systematic and that the method description is detailed*, which improves the reproducibility of our method.
>
> ---
> ***W1. (Q1) Hyperparameter Sensitivity & Backbone Generalization***
>
> We have added detailed analyses of hyperparameters to improve reproducibility and clarify the rationale behind our staged design.
>
> **1. Effect of σ₁ and σ₂**
>
> σ₁ controls low–high semantic proximity, while σ₂ regulates separation among high-level prototypes. As shown below, insufficient proximity or insufficient separation leads to performance drops. Thus, σ₁=0.25 and σ₂=1.00 provide the best balance between semantic compactness and discrimination.
>
> |  σ₁  |  σ₂  | mIoU (%) |
> |:---:|:---:|:---:|
> | 0.10 | 0.50 | 82.30  |
> | 0.10 | 1.00 | 82.37 |
> | 0.25 | 0.50 | 81.53 |
> | **0.25** | **1.00** | **84.04** |
>
> **2. Momentum coefficient m**
>
> We have evaluated how the momentum in prototype memory update affects performance. A moderate momentum (m=0.90) yields the best trade-off between stability and adaptability.
>
> | m    | mIoU (%) |
> |:----:|:---------:|
> | 0.80 | 81.34    |
> | **0.90** | **84.04**    |
> | 0.99 | 83.28    |
>
> These results confirm that the selected hyperparameters are essential for stable hierarchical prototype learning and performance improvement.
>
> **3. Backbone generalization**
>
> To address this concern, we have conducted new experiments on HRNet-W48 and SegFormer, two widely used backbones that are structurally distinct from ResNet-101. The updated full comparison is shown as follows.
>
> | Backbone    | Cityscapes | PASCAL-Part-108 | ADE20K | Mapillary |
> |:-----------:|:-----------:|:----------------:|:------:|:----------:|
> | ResNet-101 | 73.55      | 46.90           | 44.48 | 31.65     |
> | **+Ours**      | **84.04**      | **49.33**           | **48.99** | **41.42**     |
> | HRNet-W48  | 80.85      | 43.77           | 38.59 | 30.17     |
> | **+Ours**      | **81.39**      | **43.98**           | **38.95** | **43.48**     |
> | SegFormer  | 80.11      | 51.32           | 43.69 | 44.29     |
> | **+Ours**      | **80.63**      | **51.59**           | **44.29** | **45.04**     |
>
> Based on the result, we draw three key conclusions. First, our method generalizes beyond ResNet-101, yielding improvements or at least maintaining performance across all 12 backbone–dataset configurations. Second, although the gains on HRNet-W48 and SegFormer are more moderate, they are consistently positive and never negative; this is expected since these backbones already offer strong multi-scale processing (HRNet) or tokenized global representations (SegFormer), leaving relatively less room for hierarchical compression compared to ResNet-101.
>
> Finally, these results directly address the reviewer’s concern by demonstrating that the proposed method is not tied to ResNet and transfers cleanly to both CNN- and Transformer-based segmentation baselines without architectural modifications or backbone-specific tuning.
>
> ---
> ***W2. (Q1) Dense Small Objects & Severe Occlusion***
>
> In the revised paper, we have added qualitative visualizations in `Sec. 4.4` demonstrating that our method segments dense small objects and severely occluded regions accurately.

---

> > ### Author Response · Authors · 2025-12-02
> >
> > ***W3. (Q2) Deeper Comparison to SOTA***
> >
> > According to the reviewer’s concern that our Related Work lacks an in-depth contrast to SOTA methods (e.g., Contextrast [1]) and does not sufficiently explain why HiPoSeg outperforms them, **we have explicitly discussed Contextrast [1] in** `Sec. 2.3` **as a recent contrastive segmentation method** that improves optimization via context-aware sampling and hybrid objectives. In the same section, we also note a key limitation shared by many contrastive approaches: They typically assume a flat label space when defining positive/negative pairs, and thus do not explicitly encode hierarchical relations during learning.
> >
> > More importantly, the core difference is methodological. While Contextrast [1] and related contrastive methods focus on strengthening pixel representations through context- and sampling-centric designs, HiPoSeg is built around the label hierarchy itself. HiPoSeg maintains prototype memories at both high and low semantic levels and jointly optimizes them with hierarchical contrastive objectives together with a hierarchy-aware margin alignment that enforces consistent geometry across levels.
> >
> > As a result, hierarchy is not an auxiliary training trick but a representation-space organizing principle that naturally induces coarse-to-fine discrimination, while also mitigating gradient interference and prototype drift when combining multi-level objectives.
> >
> > Finally, our paper **does not merely list Contextrast [1] in tables; it provides direct comparisons on the same benchmarks.** On Cityscapes, HiPoSeg achieves **84.04%** compared to 82.20% (Contextrast [1]), and on ADE20K, HiPoSeg achieves **48.99%** compared to 43.42%. These results support that HiPoSeg’s gains stem from explicitly injecting hierarchical structure through a hierarchical prototype space and cross-level alignment, which is not targeted by flat-label contrastive frameworks.
> >
> > ---
> > ***W4. Failure Case Analysis***
> >
> > To address the concern regarding missing failure-case analysis, we have added class-wise IoU comparisons between the baseline and HiPoSeg on the Mapillary Vistas 2.0 val set at top-, high-, and low-level granularities (`Tables 9~11`), making failure patterns explicit.
> >
> > The low-level results in `Table 9` show that HiPoSeg underperforms the baseline on several categories—such as *Road Side*, *Parking*, *Service Lane*, *Traffic Island*, *Water*, and some lane-marking subclasses. We also provide an analysis of the main causes of these errors, including ambiguous boundaries and high intra-class appearance variation.
> >
> > ---
> > ***W5. (Q3) Prototype Space Evidence: Quantitative/Visualization Analysis***
> >
> > We have added `Figure 4` in `Sec. 4.4` to compare the feature spaces of the baseline and HiPoSeg. Compared to the baseline, HiPoSeg yields better class-wise feature separation while also grouping classes that share the same high-level semantics closer together, indicating that the model learns a more semantically structured embedding space.
> >
> > ---
> > [1] Sung, Changki, et al. "Contextrast: Contextual contrastive learning for semantic segmentation." Proceedings of the IEEE/CVF conference on computer vision and pattern recognition. 2024.

---

### Meta-Review · Area_Chair_7PiD · 2026-01-06

**Summary:**

The paper received initial reviews with scores of 6, 4, 4, and 4.

Reviewers shared several common concerns, including the lack of hyperparameter sensitivity analysis, issues with backbone generalization, comparisons with related work, insufficient qualitative results, and the absence of t-SNE visualizations. The authors' rebuttal adequately addressed all these issues, and the paper was revised accordingly.

Other significant issues, such as failure case analysis, fine-grained subcategory evaluation, and reliance on a fixed hierarchy tree, were all addressed in the rebuttal. However, some topics, such as validations in open-vocabulary and long-tailed settings, were considered outside the scope of the discussion and did not need to be addressed.

The area chair acknowledges that the reviewers' main concerns have been adequately addressed and therefore recommends accepting the paper.

**Reviewer Concerns:**

The reviewers' key concerns were sufficiently addressed.

**Reviewer Scores:**

None of the reviewers responded to the rebuttal before the discussion phase closed early due to the OpenReview security incident. After reviewing the review comments and the authors' responses, the area chair acknowledges that the authors have adequately addressed most of the concerns raised by the reviewers.

The reviewers likely have valid reasons for increasing their scores, with the final expected scores being **6**, **6**, **6**, and **6**.

---

### Decision · Program_Chairs · 2026-01-26

Accept (Poster)